# Improving Chain-of-Thought Efficiency for Autoregressive Image Generation

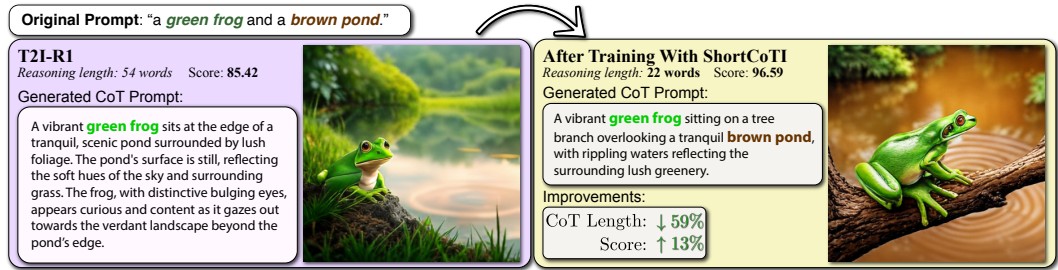

Figure 1: We observe that the reasoning CoT prompt in the T2I-R1 (Jiang et al., 2025) autoregressive image generation model often contains redundant information. To address this, we introduce ShortCoTI, the first approach aimed at improving reasoning efficiency. By incorporating a dynamic length penalty in the RL reward function, we achieve a $54\%$ improvement in reasoning efficiency on T2I-CompBench (Huang et al., 2023), as measured by token length, while also increasing accuracy by $1.14\%$. In this example, our method improves T2I-CompBench score from 85.42 to 96.59.

## Abstract

Autoregressive multimodal large language models have recently gained popularity for image generation, driven by advances in foundation models. To enhance alignment and detail, newer approaches employ chain-of-thought (CoT) reasoning, expanding user inputs into elaborated prompts prior to image synthesis. However, this strategy can introduce unnecessary redundancy—a phenomenon we call *visual overthinking*—which increases computational costs and can introduce details that contradict the original prompt. In this work, we explore how to generate more concise CoT sequences for more efficient image generation. We introduce **ShortCoTI**, a lightweight optimization framework that encourages more concise CoT while preserving output image quality. ShortCoTI rewards more concise prompts with an adaptive function that scales according to an estimated difficulty for each task. Incorporating this reward into a reinforcement learning paradigm reduces prompt reasoning length by $54\%$ while maintaining or slightly improving quality metrics across multiple benchmarks (T2I-CompBench, GenEval). Qualitative analysis shows that our method eliminates verbose explanations and repetitive refinements, producing reasoning prompts that are both concise and semantically rich. As a result, ShortCoTI improves computational efficiency without compromising the fidelity or visual appeal of generated images.

## 1 Introduction

The rapid advancement of multimodal foundation models has transformed generative AI, driving remarkable progress in text-to-image generation. These models are now capable of translating textual prompts into increasingly sophisticated visual outputs. A crucial yet under-explored aspect of this process is the model's internal chain-of-thought (CoT) mechanism, where user inputs are reasoned through and automatically expanded into more detailed, and sometimes multi-internal-step descriptions before generating an image (see Fig. 1 for an example). Notable works in this area include Bagel (Deng et al., 2025), T2I-R1 (Jiang et al., 2025), ReasonGen-R1 (Zhang et al., 2025),

and ImageGen-CoT (Guo et al., 2025; Liao et al., 2025). In parallel, modern standalone diffusion models such as Stable Diffusion XL (Podell et al., 2023), DALL·E 3 (Betker et al., 2023), and Emu (Dai et al., 2023) utilize advanced prompt rewriting systems powered by large language models (LLMs) to analyze and elaborate user input before image generation, effectively mimicking a reasoning process.

While this reasoning often enhances output quality, it can also introduce inefficiencies: each additional token in the reasoning prompt increases computational cost, echoing the well-documented "overthinking" phenomenon in reasoning LLMs (Sui et al., 2025). For example, as illustrated by the ball-party hat scenario in Fig. 2, verbose CoT reasoning can add redundant descriptions that are unnecessary for generating the desired objects. Removing these superfluous details, even manually, can maintain generation quality (columns (b)-(d)) while improving efficiency. This observation motivates our study of reasoning efficiency in autoregressive image generation.

There are significant amounts of recent work in the LLM field to improve the efficiency of CoT. Liu et al. (2025) and Sui et al. (2025) provide good survey summaries of SotA methods. Methods to improve the efficiency of LLM reasoning include, but are not limited to, using RL (Wu et al., 2025; Yi et al., 2025; Arora & Zanette, 2025), SFT with variable data length (Xia et al., 2025), prompt-level control (Han et al., 2024), dynamic budgeting (Huang et al., 2025; Li et al., 2025b), and using a router to assign different tasks to different reasoning "modes" (Ong et al., 2024).

On the other hand, addressing CoT redundancy in autoregressive image generation introduces unique challenges that set it apart from LLMs. First, unlike text-only tasks where output quality can be directly evaluated by answer correctness, image generation demands careful preservation of nuanced alignment between the user's textual intent and the resulting visual output. Second, the relationship between input prompt length and image quality is highly nonlinear: while some concise prompts yield excellent images, others require elaborate reasoning to achieve the desired fidelity and detail.

In this paper, we study methods for improving the efficiency of chain-of-thought (CoT) reasoning in autoregressive image generation and propose three strategies for reducing CoT length: Cap Length, Target Length and ShortCoTI. All three approaches substantially shorten CoTs, with the most effective being **ShortCoTI**. With ShortCoTI, we introduce a reinforcement learning system based on Group Relative Policy Optimization (Shao et al., 2024) that dynamically optimizes prompt length while preserving both visual fidelity and text alignment. Our approach incorporates a length penalty loss to encourage shorter reasoning and a reward model to promote accuracy. By extending LLM-based efficiency methods to handle the alignment constraints unique to autoregressive image generation, our method bridges the gap between text and visual domains. Fig. 1 shows an example. In summary, our contributions are:

- We show that "overthinking" redundancy is not unique to LLM text tasks. It is also prevalent in current autoregressive image-generation CoT methods, as observed in T2I-R1.

- To address both the necessity and conciseness of CoT reasoning, we develop ShortCoTI, a reinforcement learning algorithm based on GRPO with accuracy rewards and a CoT length penalty, which adaptively reduces CoT length.

- Using T2I-R1 as our base model, we demonstrate that our methods significantly improve image generation efficiency. ShortCoTI reduces reasoning token length by $54\%$ while improving generation quality by $1.44\%$ on T2I-CompBench (Huang et al., 2023) and $2.76\%$ on GenEval (Ghosh et al., 2023).

## 2 RELATED WORK

### 2.1 CHAIN-OF-THOUGHT EFFICIENCY IN LLM

Chain-of-Thought (CoT) reasoning has emerged as a fundamental technique for enabling large language models (LLMs) to tackle complex tasks through step-by-step problem decomposition (Wei et al., 2022). Despite its effectiveness, CoT reasoning often suffers from the "overthinking" phenomenon, where models produce unnecessarily verbose or redundant reasoning chains. This inefficiency has motivated a growing body of research aimed at optimizing CoT for both brevity and accuracy (Liu et al., 2025; Sui et al., 2025).

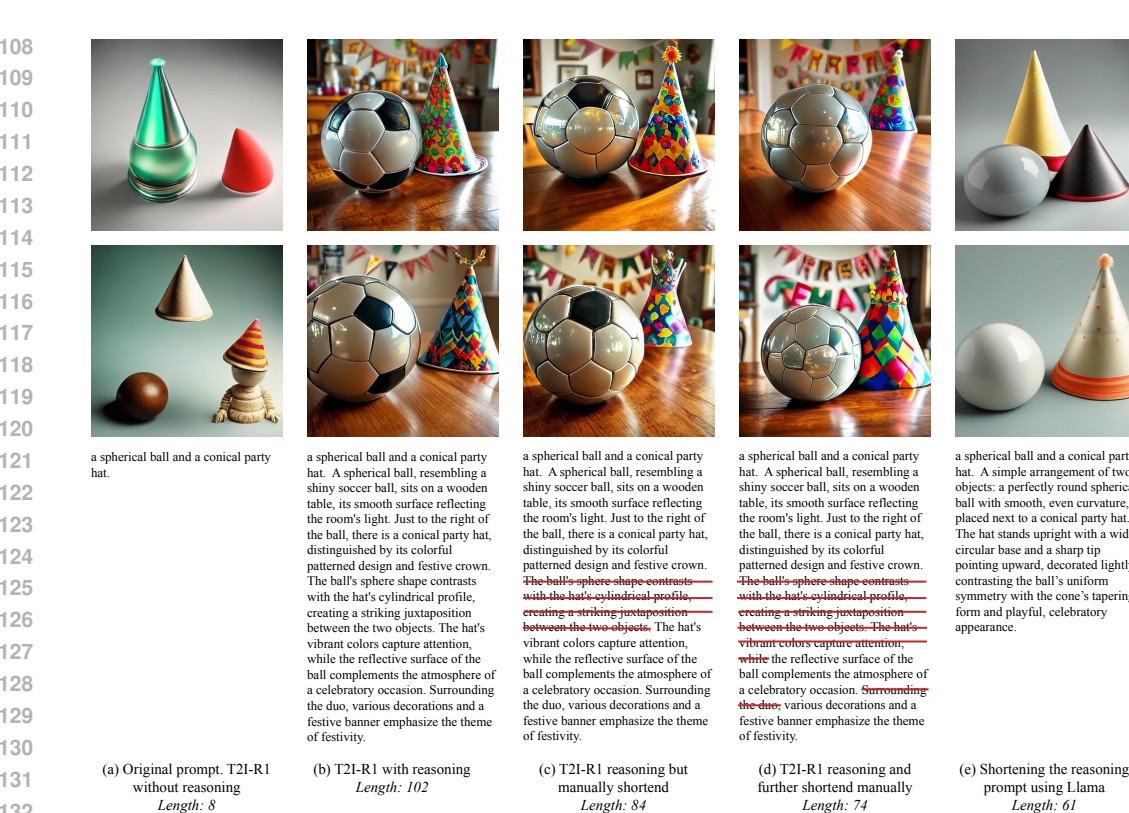

a spherical ball and a conical party hat.

a spherical ball and a conical party hat. A spherical ball, resembling a shiny soccer ball, sits on a wooden table, its smooth surface reflecting the room's light. Just to the right of the ball, there is a conical party hat, distinguished by its colorful patterned design and festive crown. The ball's sphere shape contrasts with the hat's cylindrical profile, creating a striking juxtaposition between the two objects. The hat's vibrant colors capture attention, while the reflective surface of the ball complements the atmosphere of a celebratory occasion. Surrounding the duo, various decorations and a festive banner emphasize the theme of festivity.

a spherical ball and a conical party hat. A spherical ball, resembling a shiny soccer ball, sits on a wooden table, its smooth surface reflecting the room's light. Just to the right of the ball, there is a conical party hat, distinguished by its colorful patterned design and festive crown. ~~The ball's sphere shape contrasts with the hat's cylindrical profile, creating a striking juxtaposition between the two objects.~~ The hat's vibrant colors capture attention, while the reflective surface of the ball complements the atmosphere of a celebratory occasion. Surrounding the duo, various decorations and a festive banner emphasize the theme of festivity.

a spherical ball and a conical party hat. A spherical ball, resembling a shiny soccer ball, sits on a wooden table, its smooth surface reflecting the room's light. Just to the right of the ball, there is a conical party hat, distinguished by its colorful patterned design and festive crown. ~~The ball's sphere shape contrasts with the hat's cylindrical profile, creating a striking juxtaposition between the two objects. The hat's vibrant colors capture attention, while~~ the reflective surface of the ball complements the atmosphere of a celebratory occasion. ~~Surrounding the duo,~~ various decorations and a festive banner emphasize the theme of festivity.

a spherical ball and a conical party hat. A simple arrangement of two objects: a perfectly round spherical ball with smooth, even curvature, placed next to a conical party hat. The hat stands upright with a wide circular base and a sharp tip pointing upward, decorated lightly, contrasting the ball's uniform symmetry with the cone's tapering form and playful, celebratory appearance.

| (a) Original prompt. T2I-R1 without reasoning *Length: 8* | (b) T2I-R1 with reasoning *Length: 102* | (c) T2I-R1 reasoning but manually shortend *Length: 84* | (d) T2I-R1 reasoning and further shortend manually *Length: 74* | (e) Shortening the reasoning prompt using Llama *Length: 61* |

Figure 2: We began by investigating whether we can manually shorten the reasoning prompt while maintaining the quality of images generated by T2I-R1 (Jiang et al., 2025). We found that we can successfully delete unnecessary sentences in the reasoning prompt while maintaining the generation quality in many cases (columns (c) and (d), where red lines shows what we delete with respect to the original CoT in (b)). However, using an off-the-shelf LLM such as Llama (Dubey et al., 2024) to shorten the reasoning prompt often cannot maintain the key information useful for image generation, thus leading to a degradation of the image generation quality. This motivates us to post-train the model end-to-end to more intelligently improve CoT efficiency.

A comprehensive survey of LLM reasoning efficiency is beyond the scope of this section. Instead, we highlight several representative approaches: Model-based methods such as ARM (Wu et al., 2025), ShorterBetter (Yi et al., 2025), and the approach by (Arora & Zanette, 2025) leverage adaptive format selection or reinforcement learning with length penalties to minimize token usage while maintaining performance. Xia et al. (2025) employ supervised finetuning with length-controlled CoT data to improve token compression rates. Dynamic budgeting strategies have also been proposed. AdaCtrl (Huang et al., 2025) and SelfBudgeter (Li et al., 2025b) introduce mechanisms that allocate reasoning budgets based on problem difficulty. Han et al. (2024) utilize prompt-level controls to constrain token budgets, while RouteLLM (Ong et al., 2024) explores routing to different LLMs according to task complexity. Additionally, AALC (Li et al., 2025a) seeks to balance accuracy and brevity in generated outputs, though its reward functions (e.g., answer correctness) are not directly applicable to multimodal tasks. While these methods have demonstrated success in language-based reasoning, their extension to image generation domains remains an open research question.

## 2.2 CHAIN-OF-THOUGHT IN IMAGE GENERATION

Recent work has begun to integrate Chain-of-Thought (CoT) reasoning into image generation. ReasonGen-R1 (Zhang et al., 2025) uses reinforcement learning to generate textual rationales before image synthesis, while ImageGen-CoT (Guo et al., 2025) enhances in-context learning by requiring explicit reasoning traces. T2I-R1 (Jiang et al., 2025) introduces bi-level CoT, combining semantic planning with token-level patch generation. While these methods show that CoT can improve image

quality, they inherit a key limitation from early LLM CoT work: the assumption that longer reasoning is preferred, leading to inefficiency. To our knowledge, no prior work addresses redundancy or efficiency in visual CoT steps.

Our work is the first to bridge LLM-style CoT efficiency with image generation. Unlike previous methods that focus solely on quality, we identify and quantify overthinking in visual CoT, and adapt LLM efficiency strategies to the multimodal image generation domain.

## 3 METHOD

We use T2I-R1 (Jiang et al., 2025) as the base model for our experiments. In this section, we first provide a brief overview of T2I-R1 and its reinforcement learning procedure. We then present our proposed method, along with several alternative approaches, to enhance reasoning efficiency.

### 3.1 PRELIMINARIES

Autoregressive image generation models produce images token by token, conditioning each step on the prompt and prior tokens. T2I-R1 enhances autoregressive text-to-image generation with a two-stage Chain-of-Thought (CoT) process: a semantic-level CoT first constructs a high-level plan in text, followed by a token-level CoT that generates image tokens to realize it. Trained with BiCoT-GRPO (Equation 1), this collaborative CoT approach yields images that are more coherent, aligned, and diverse than the baseline method.

$$\mathcal{J}_{\text{GRPO}}(\theta) = \mathbb{E}_{\substack{(q,a) \sim \mathcal{D}, \\ \{o_i\}_{i=1}^{G} \sim \pi_{\theta_{\text{old}}}(\cdot|q)}} \left[ \frac{1}{\sum_{i=1}^{G} |o_i|} \sum_{i=1}^{G} \sum_{t=1}^{|o_i|} \Big( \min\big(r_{i,t}(\theta)\hat{A}_i, \right.$$
$$\left. \text{clip}(r_{i,t}(\theta), 1 - \varepsilon, 1 + \varepsilon)\hat{A}_i\big)\Big) \right] - \beta\, D_{\text{KL}}(\pi_\theta \,\|\, \pi_{\text{ref}}) \tag{1}$$

where

$$A_i = \frac{\mathcal{R}_i - \text{mean}(\{\mathcal{R}_i\}_{i=1}^{G})}{\text{std}(\{\mathcal{R}_i\}_{i=1}^{G})} \tag{2}$$

and

$$r_{i,j}(\theta) = \frac{\pi_\theta(o_{i,j}|q, o_{i,<j})}{\pi_{\theta_{\text{old}}}(o_{i,j}|q, o_{i,<j})} = \begin{cases} \frac{\pi_\theta(s_{i,j}|q,s_{i,<j})}{\pi_{\theta_{\text{old}}}(s_{i,j}|q,s_{i,<j})}, & 0 \le j \le |s_i| \\ \frac{\pi_\theta(t_{i,j}|q,s_i,t_{i,<j})}{\pi_{\theta_{\text{old}}}(t_{i,j}|q,s_i,t_{i,<j})}, & |s_i| < j \le |s_i| + M \end{cases} \tag{3}$$

Here, $(p, a)$ are prompt and ground truth, $G$ is a group of individual responses. $\{o_i\}_{i=1}^{G}$ is sampled from the old policy $\pi_{\theta_{\text{old}}}$. $\mathcal{R}_i$ is the individual reward. $A_i$ is interpreted as the advantage of the $i$-th response is calculated by normalizing the rewards $\{\mathcal{R}_i\}_{i=1}^{G}$ of the group. The main novelty of T2I-R1 is the $r_{i,j}(\theta)$ term, where $s_i$ is the sematic level CoT composed of $|si|$ text tokens of $\{s_{i,1}, s_{i,2}, \cdots, s_{i,|si|}\}$ and $t_i$ consists of $M$ image tokens $\{t_{i,1}, t_{i,2}, \cdots, t_{i,M}\}$, and $o$ is the response that consists of both semantic and image tokens, $o_i = (s_i, t_i)$.

T2I-R1 employs an "ensemble of generation rewards", combining a human preference model, an object detector, a VQA model, and an output reward model to form the final reward signal during GRPO training. Integrating these four reward models helps mitigate the "reward hacking" often seen in reinforcement learning, thereby enabling more effective CoT reasoning.

### 3.2 OVERALL METHOD

Our method introduces a simple yet effective modification to T2I-R1's reward functions by incorporating a length penalty that dynamically adapts to task difficulty. Specifically, given the original rewards, we define a new reward function:

$$R_{ShortCoTI} = -\alpha * f(R_{models}) * L(y) \tag{4}$$

where $f$ is a scaling function over the baseline rewards, y is the rewritten prompt (CoT) used for image generation, $L(y)$ measures its length, and $\alpha$ controls the penalty's strength.

One key difference of applying RL on image generation versus some LLM reasoning tasks is the former's rewards are often not verifiable or binary. Previous works focusing on reducing CoT length for text generation often use binary reward to determine the difficulty of the problem. However, we still want to use the other rewards in T2I-R1 on object detection, human preference, question answering, etc. to estimate the difficulty of each prompt to the actor model. A native way is to binarize the reward with certain "hard" thresholds, i.e. set $f(R_{models}) = 1$ if $R_{models_i} > t_i \forall i \in [0, len(models)]$, and 0 otherwise. Another option is to design the length penalty to be proportional to the raw summation of those rewards. As the rewards used by T2I-R1 include GIT (Wang et al., 2022), GroundingDINO (Liu et al., 2024) and HPSv2 (Wu et al., 2023), which are concentrated on the range of $[0.2, 0.8], [0.6, 1.0], [0.26, 0.32]$, respectively, the sum is around $[1.06, 2.12]$. We set $f(R_{models}) = R_{models} - 1$ to offset the baseline. We refer to the hard and soft versions as ShortCoTI (hard) and ShortCoTI (soft), respectively, in the following sections.

Intuitively, when alignment or fidelity rewards are high (indicating an easier task), we apply a stronger penalty to discourage unnecessarily long sequences. For harder tasks (lower rewards), the penalty is relaxed, allowing for more detailed reasoning. This adaptive weighting aggressively trims redundancy in simple cases while preserving flexibility for complex reasoning when needed.

### 3.3 ALTERNATIVE LENGTH PENALTY METHODS

Inspired by efficiency techniques in language CoT, we further explore two strategies for reducing the length of image generation CoT: Cap Length and Target Length.

**Cap Length** The CoT is forcibly shortened by discarding tokens beyond a fixed limit. These excess tokens are omitted from the image generation stage, enforcing a hard cutoff regardless of semantic completeness. Note that all strategies not done only during test time, but rather, during the fine-tuning. Therefore, it could force the model to pay more attention to the original prompt and the short CoT, even if truncation causes imcompleteness.

**Target Length:** This approach enforces a fixed target length $L_T$:

$$R_{TargetLength} = -\alpha * max(0, L(y) - L_T) \tag{5}$$

assigning a linear penalty for any deviation from the target. This could be interpreted as ShortCoTI without dependence on prompt difficulty, with $L_T$ adjusting the "harshness" by providing a target length. Tab. 1 shows a summary of all proposed methods.

Table 1: Summary of CoT shortening methods and descriptions.

| Method | Description |
| --- | --- |
| Cap Length | Truncate the reasoning prompt after $N$ words |
| Target Length | Length penalty given a target reasoning length $L_T$ |
| ShortCoTI (hard) | Length penalty loss with binarized penalty depending on prompt difficulty |
| ShortCoTI (soft) | Length penalty loss with soft penalty depending on prompt difficulty |

## 4 EXPERIMENTS

### 4.1 IMPLEMENTATION DETAILS

We build upon T2I-R1, intentionally keeping the setups as unchanged as possible to isolate the effects of the new reward function on CoT length. We use T2I-R1's base model and reward functions (Jiang et al., 2025), as well as its template to prompt the model for CoT (details in Sec. 5). The learning rate is $1e^{-6}$ and the batch size for each GPU is 1. We train all our methods (including the baselines) for 800 epochs. We set the $\alpha$ coefficient for the ShortCoTI version of our length reward to $5e^{-4}$, and for other variations, we set the coefficients such that the initial magnitude of the reward is close to that of ShortCoTI (soft), ensuring fairness in the impact of our component to the training process. Specifically, the $\alpha$ for ShortCoTI (hard) and Target Length are $1e^{-3}$ and $5e^{-4}$ respectively. For $L_T$, we set it for Cap Length and Target Length to be close to the stablized length of ShortCoTI (soft), which is 35 tokens. Please see the supplemental for more details on training statistics.

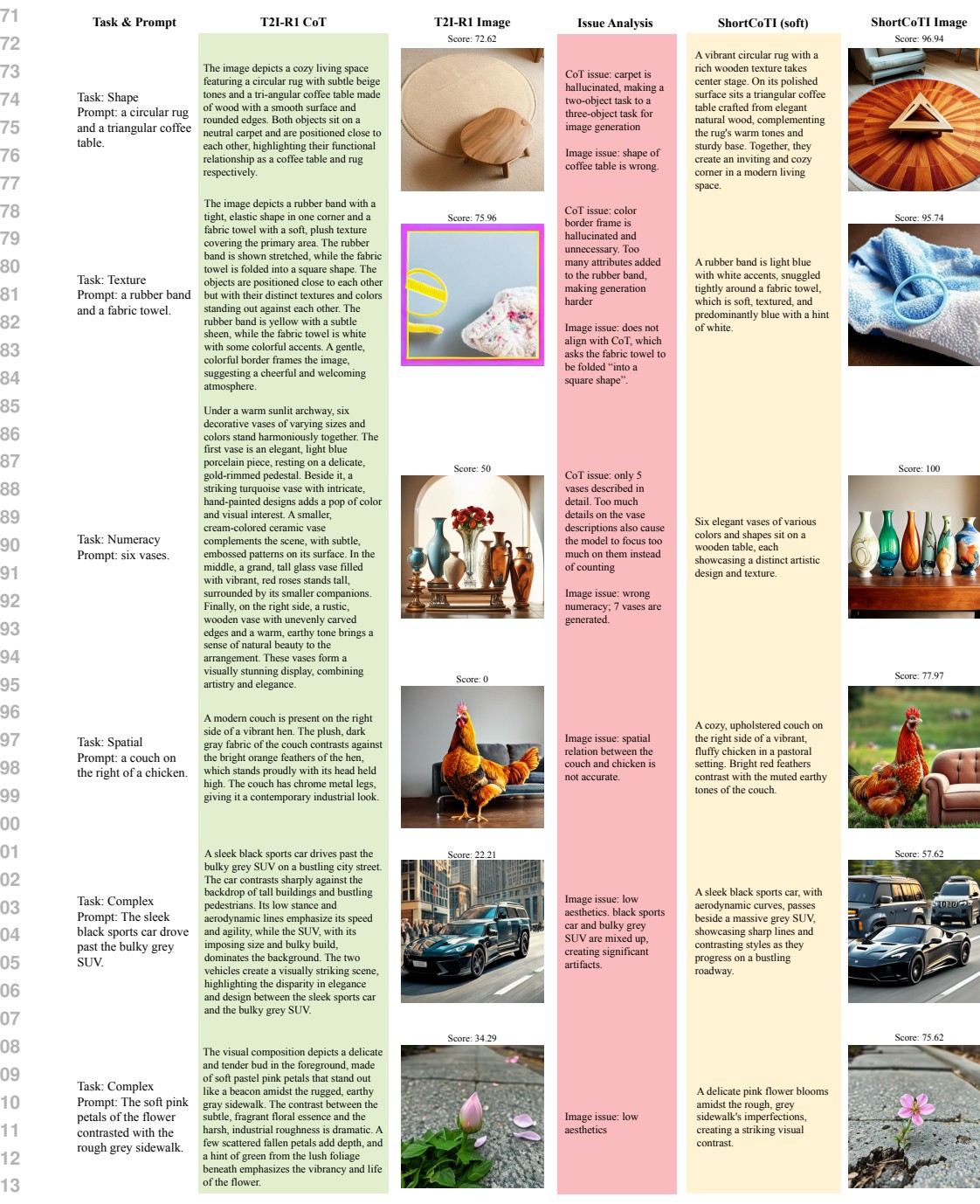

Figure 3: Trained with reward functions that combines generation accuracy and length penalty, our model achieves concise CoT while preserving, or even improving image quality in some cases. In rows 1-3, baseline T2I-R1 generate hallucinated or incorrect objects and details in the reasoning prompt. Excessive and irrelevant content increases generation difficulty, causing the model to overlook or misrepresent the desired objects and attributes specified in the prompt. In rows 4–6, we show that our training approach enhances the model's overall ability to follow prompts. Even when the baseline T2I-R1's CoT does not contain obvious errors, our model achieves higher prompt accuracy and improved image quality. The scores above images are evaluated by the judge models in T2I-CompBench, which correspond to our visual findings. All results are generated with the same seed.

## 4.2 EVALUATION DATASETS

We evaluate our model on two publicly available benchmarks: GenEval (Ghosh et al., 2023) and T2I-CompBench (Huang et al., 2023). GenEval is a text-to-image benchmark designed to assess how well generative models follow compositional instructions, with an emphasis on fine-grained alignment between text prompts and generated images. It tests models using controlled prompts across tasks such as object shape, color, texture, counting, spatial relations (2D/3D), non-spatial relations, and compositional complexity. Similarly, T2I-CompBench also evaluates text-to-image models on compositional generalization, measuring how accurately models generate images that reflect multiple attributes and relationships described in prompts. It features more diverse prompt types covering object properties, spatial relations, numeracy, and complex multi-object compositions, and is generally more challenging than GenEval.

## 4.3 COMPUTATION COST

Our method significantly reduces the average prompt plus CoT length, cutting it from 93.11 tokens in the baseline to 41.97 tokens, a reduction of approximately 54.9%, measured on T2I-R1 dataset, shown in Tab. 2. We also report end-to-end image generation time. The majority of time consumption is for the autoregressive image generation itself, so this reduces the total inference time by 8.14%. Excluding the image generation time which averages around 29.48 seconds, the pure runtime improvement of the reasoning part is 52.88%. Efficiency improvement in the image generation model (such as doing distillation or using a diffusion head) is beyond the scope of this paper.

Table 2: Computational Effeciency Comparison. Length is measured by number of words.

| Task | w/o CoT | T2I-R1 | ShortCoTI (hard) | ShortCoTI (soft) |
|---|---|---|---|---|
| CoT Length | 0 | 93.11 | 45.21 | **41.97** |
| Inference Time (s) | 29.48 | 34.85 | 32.22 | **32.01** |

## 4.4 TEXT-TO-IMAGE ALIGNMENT

Table 3: Text-to-Image Alignment Comparison on GenEval.

| Metric | Single Object | Counting | Two Objects | Position | Color_Attr | Colors | Overall |
|---|---|---|---|---|---|---|---|
| T2I-R1 | 98.75 | 54.06 | **92.42** | 75.25 | 62.88 | 87.77 | 78.52 |
| Trunc | 99.69 | 51.56 | 90.91 | 82.75 | 63.89 | 89.10 | 79.65 |
| Target Length | **99.69** | 55.31 | 90.91 | 80.25 | **66.41** | 86.70 | 79.87 |
| ShortCoTI (hard) | 99.38 | 53.75 | 90.15 | 82.50 | 61.62 | 89.10 | 79.41 |
| ShortCoTI (soft) | 99.06 | **55.62** | 91.67 | **83.75** | 64.14 | **89.89** | **80.69** |

Table 4: Text-to-Image Alignment Comparison on T2I-CompBench.

| Task | Shape | Color | Texture | Numeracy | 2D Spatial | 3D Spatial | Non-Spatial | Complex | Overall |
|---|---|---|---|---|---|---|---|---|---|
| T2I-R1 | 60.06 | 82.46 | 73.21 | **62.43** | 32.09 | **39.36** | 30.96 | 40.33 | 52.61 |
| Cap Length | 59.45 | 83.02 | **75.68** | 59.19 | 35.35 | 38.12 | 31.06 | 40.45 | 52.79 |
| Target Length | 59.99 | **84.29** | 74.76 | 59.96 | 33.62 | 38.95 | 31.18 | 40.16 | 52.86 |
| ShortCoTI (hard) | 58.36 | 83.47 | 74.75 | 61.44 | 34.92 | 38.31 | **31.18** | **40.56** | 52.87 |
| ShortCoTI (soft) | **60.40** | 83.56 | 74.76 | 60.64 | **38.12** | 38.16 | 31.06 | 40.32 | **53.37** |

The results presented in Tab. 3 and Tab. 4 indicate that our method outperforms T2I-R1 (Jiang et al., 2025) in accuracy on the GenEval (Ghosh et al., 2023) and T2I-CompBench (Huang et al., 2023) benchmarks. We see especially significant improvement in "position" and "2D spatial relationship" tasks. While all our proposed methods are strong, in general, the best candidate is ShortCoTI (soft). These findings support our hypothesis that more efficient reasoning with adjustment based on prompt difficulties not only decreases generation latency but also enhances text-image alignment accuracy.

Fig. 3 provides a qualitative comparison of CoT outputs from T2I-R1 and our ShortCoTI (soft) method across a range of text-to-image tasks. Our method addresses common failure cases and hallucinations in the CoT prompt, such as miscounting, inaccurate spatial relationships, and suboptimal aesthetics (rows 1-3), thus producing images that are more faithful to the prompts and exhibit greater visual coherence. We also observe that even in cases where the original T2I-R1's long CoT reasoning do not have obvious errors (rows 4-6), our model still outperforms T2I-R1 in generation accuracy. We hypothesis that 1) concise CoT reduces the number of unnecessary text tokens, so the image generation step can focus its capacity more on the important objects and attributes with fewer text tokens to attend, and 2) the CoT-length-constrained reinforcement learning inherently improves the dynamics of the model and improves the generation quality.

## 4.5 AESTHETICS

We further validate that our proposed methods do not compromise the visual aesthetics of generated images. To assess the perceptual quality of the outputs, we use an aesthetic predictor similar to Laion-Aesthetics (Schuhmann et al., 2022). Specifically, we extract image embeddings and input them into a lightweight linear regression head, where higher scores indicate stronger alignment with human aesthetic judgments. This automatic metric enables us to compare prompt-rewriting strategies not only in terms of efficiency and alignment, but also with respect to visual appeal.

Tabs. 5 and 6 present aesthetic scores on GenEval (Ghosh et al., 2023) and T2I-CompBench (Huang et al., 2023). The aesthetic scores for all methods are comparable, indicating that our approach largely preserves the aesthetic quality of outputs. Cap Length, which generally uses the fewest tokens, has slightly higher scores overall, which may suggest that less specific prompting has a tendency to yield higher aesthetic scores. This effect is small, but could reflect a bias towards aesthetically pleasing images in the generator's training data.

Table 5: Aesthetics Comparison on GenEval.

| Method | Single Obj | Counting | Two Objs | Position | Color_Attr | Colors | Overall |
|---|---|---|---|---|---|---|---|
| T2I-R1 | 6.25 | **5.86** | 6.20 | 6.03 | 6.25 | 6.42 | 6.17 |
| Cap Length | **6.30** | **5.86** | **6.22** | 6.05 | **6.34** | **6.48** | **6.21** |
| Target Length | 6.27 | 5.79 | 6.15 | 6.03 | 6.24 | 6.45 | 6.16 |
| ShortCoTI (hard) | 6.18 | 5.82 | 6.16 | **6.06** | 6.26 | 6.42 | 6.16 |
| ShortCoTI (soft) | 6.27 | 5.82 | 6.21 | 6.00 | 6.25 | 6.42 | 6.17 |

Table 6: Aesthetics Comparison on T2I-CompBench.

| Task | Shape | Color | Texture | Numeracy | 2D Spatial | 3D Spatial | Non-Spatial | Complex | Overall |
|---|---|---|---|---|---|---|---|---|---|
| T2I-R1 | 5.94 | 6.46 | 5.82 | 6.23 | **6.25** | 6.07 | 6.50 | **5.89** | 6.14 |
| Cap Length | **5.99** | **6.49** | **5.91** | 6.23 | **6.25** | 6.09 | 6.50 | 5.85 | **6.16** |
| Target Length | 5.88 | 6.46 | 5.79 | 6.22 | 6.16 | 6.02 | **6.55** | 5.86 | 6.11 |
| ShortCoTI (hard) | 5.95 | 6.47 | 5.74 | 6.20 | 6.17 | **6.13** | 6.51 | 5.86 | 6.12 |
| ShortCoTI (soft) | 5.96 | 6.37 | 5.78 | **6.26** | 6.20 | 6.07 | 6.47 | 5.85 | 6.12 |

## 5 DISCUSSION

We first visualize the CoT length distribution of ShortCoTI. As shown in Fig. 4(a), it resembles a right-skewed Gaussian distribution with mean around 40. In the following, we will use the shape subtask of T2I-CompBench, comprising of 300 prompts in the validation set, to further analyze our result the influence of CoT seeds, prompting template.

**Prompting Templates** For our main results, we stick to T2I-R1's template to prompt the base model for CoT from the original input, which asks for a CoT to "1. Include every object mentioned in the prompt; 2. Specify visual attributes (color, number, shape, texture) if specified in the prompt; 3. Clarify relationships (e.g., spatial) between objects if specified in the prompt ..." However, there are many other possible templates, and we can think of one template as guiding reasoning in a

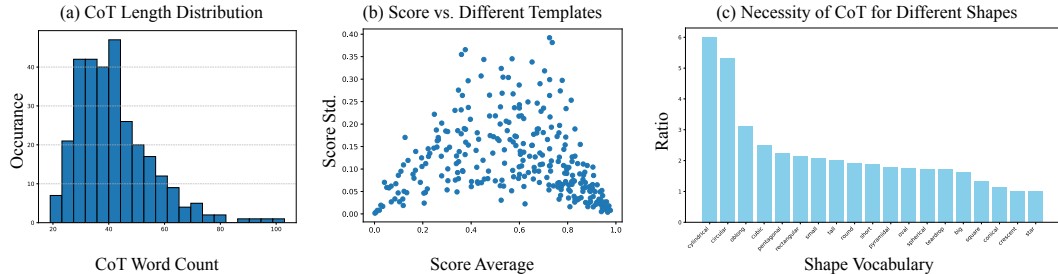

Figure 4: (a) Distribution of CoT length. (b) We draw the distribution of different scores for the standard deviation of the generation CoT length across 4 different prompt template for inference-time scaling. With more template to choose from, the score slightly improves. (c) we estimate the difficulty of the shape words in shape subtask of T2I-CompBench by the effects of CoT.

certain logic order. For example, it could also ask the model to describe objects from the general background to the more detailed foregrounds. We also use 3 additional templates, and along with the original one, gathering 4 results for each input prompt. The average and standard deviation of scores are plotted in Fig. 4(b). An interesting phenomenon is that when the image scores are very high or low, i.e. the prompt is very easy or hard, the Std. is low, suggesting different "way of thinking" does not have much effects. However, when the task difficulty is medium, different "way of thinking" can lead to very different results. This indicates test-time scaling would be helpful in this dimension, and we leave it as future work. We also discuss other potential future directions in the Appendix.

**Necessity of CoT** Another interesting question is whether CoT is always helpful for all input prompts. By generating a comparison set of results with CoT disabled, we found the answer is negative. Sometimes no CoT gives better text-image alignment. We hypothesize prompts that need CoT are more difficult, and briefly investigated the shape adjectives in the shape subtask. For each shape, we count the number of prompts having a better result with CoT than without, and calculate the ratio of these two occurances. The ratio thus indicates how likely prompts with such shapes would benefit from CoT. As shown in Fig. 4(c), the top ones are more complicated shapes such as "cynlindrical", "pentagonal", and the simpler ones such as "short", "big" have smaller ratios.

**Seeds** We generate results with four seeds and calculate the average and standard deviation (Std.) of scores and CoT lengths, acquiring Pearson correlation coefficient in Tab. 7. The strongest positive correlation is between CoT length mean and Std., meaning longer CoT has larger length variance. The strongest negative correlation is between the mean score and the mean CoT length, suggesting our method has successfully captured the difficulty dependency to certain extent.

Table 7: Pearson Correlation Coefficient. As the matrix is symmetric, we omit the bottom half.

|  | Length_AVG | Length_Std. | Score_AVG | Score_Std. |
|---|---|---|---|---|
| Length_AVG | 1.00 | 0.52 | -0.21 | -0.11 |
| Length_Std. | - | 1.00 | 0.05 | 0.02 |
| Score_AVG | - | - | 1.00 | -0.20 |
| Score_Std. | - | - | - | 1.00 |

# 6 CONCLUSION

In this work, we took the first step towards efficient CoT reasoning for autoregressive image generation. We identified frequent redundancies in current prompt expansion stage of text-to-image generation, and designed ShortCoTI, a lightweight optimization framework, to efficiently shorten visual CoT sequences while maintaining high output quality. By incorporating an difficulty-aware reward and adaptive length penalties within a reinforcement learning framework, ShortCoTI successfully reduces prompt rewriting length by 54% on T2I-CompBench and GenEval, all without sacrificing image-text alignment or fidelity. We believe this first step towards autoregressive image generation efficiency work opens door for many other follow-up application.

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

# A    SUPPLEMENTAL MATERIALS

## A.1    TRAINING STATISTICS

We provide curves of our reward function values and the overall CoT length change across training steps in Fig. 5. We train with 4 rollouts during the first 600 epochs, and then lowered to 3 for the last 200. Cap Length does not have length reward and its CoT length is fixed at 35 tokens.

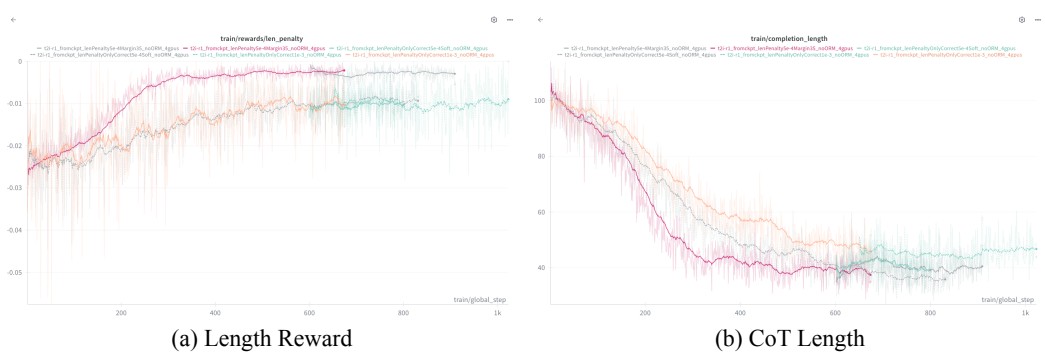

(a) Length Reward                                                  (b) CoT Length

Figure 5: Training Statistics of Our 4 Strategies.

## A.2    PROMPTING TEMPLATES

Given an input prompt, the templates we use to prompt the base model to generate CoT are shown in Fig. 6. We use (a) for all main results, and explore (b)-(d) in Sec. 5.

#### (a) Original Template

You are asked to generate an image based on this prompt: "{}"
Provide a brief, precise visualization of all elements in the prompt. Your description should:
1. Include every object mentioned in the prompt
2. Specify visual attributes (color, number, shape, texture) if specified in the prompt
3. Clarify relationships (e.g., spatial) between objects if specified in the prompt
4. Be concise (50 words or less)
5. Focus only on what's explicitly stated in the prompt
6. Do not elaborate beyond the attributes or relationships specified in the prompt
Do not miss objects. Output your visualization directly without explanation:

#### (b) Extra Template 1

You are asked to generate an image based on this prompt: "{}"
Think step by step. First, identify all objects mentioned.
Next, specify each object's explicit attributes (color, number, shape, texture).
Finally, describe the spatial or relational connections between them. Ensure nothing is omitted.
Think about your reasoning first, then answer with a single coherent visualization description. Use less than 50 words for the final description. DO NOT output your thinking process.
Output your visualization directly without explanation:

#### (c) Extra Template 2

You are asked to generate an image based on this prompt: "{}"
Imagine to decompose the scene into layers:
1. Overall environment or background if stated.
2. Foreground and main objects.
3. Secondary objects or supporting details. AVOID unnecessary details that is NOT in the prompt.
For each object, imagine its attributes and relationships precisely. Reply with one concise scene description suitable for image generation. Use less than 50 words. DO NOT output your thinking process. Output your visualization directly without explanation:

#### (d) Extra Template 3

You are asked to generate an image based on this prompt: "{}"
Organize reasoning in a structured way: make an implicit table where each object is a row, and its columns are attributes (color, number, shape, texture) and spatial relationships.
Ensure every detail is filled if given.
Then convert this structured reasoning into a single compact visualization description. Use less than 50 words. DO NOT output your thinking process. Output your visualization directly without explanation, with the format: "Visualization: <your answer>".

Figure 6: Prompting Templates.

### A.3 MORE RESULTS

In Fig. 7 we provide more (uncurated) result visualization from Cap Length and Target Length. As can be seen from the CoTs of top 2 rows, Cap Length does not learn to make sentences complete within the cut-off budget. However, as we optimize the image and CoT tokens end-to-end, the model is able to use incomplete CoT to still generate plausible images.

Task: Two Objects
Prompt: a photo of a toothbrush and a bench.

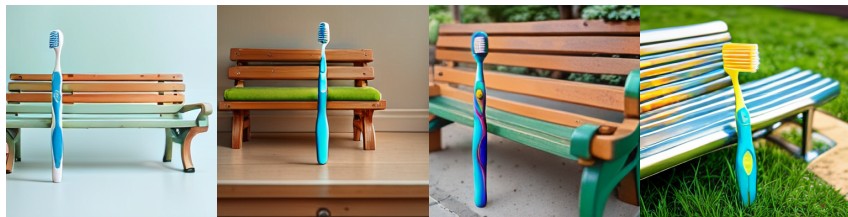

A vibrant photograph depicting a modern toothbrush with a soft white and blue handle, standing beside a quaint wooden bench painted in gentle pastel greens. The toothbrush's bristles

This image showcases a toothbrush standing upright on a wooden bench with a green cushion for comfort. The toothbrush is white with blue accents, and a vibrant color scheme. The

A toothbrush with a colorful design on the bristles and a wooden bench painted in shades of green is visible in the image. The bench is situated near a small park,

A vibrant outdoor setting features a vivid yellow and blue toothbrush sitting atop green grass beside a sleek metallic bench with smooth contours and reflective surfaces. Together, they create a cheerful and

Task: Counting
Prompt: a photo of two parking meters.

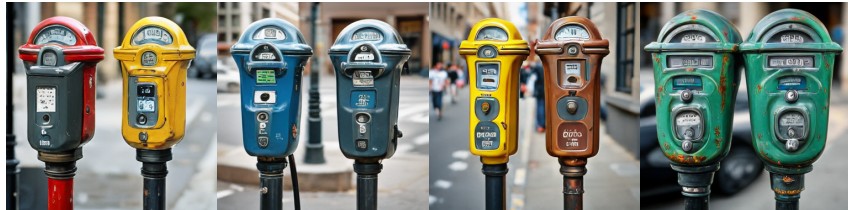

A classic image showing two old-fashioned parking meters, one painted red and the other yellow, side by side on a street corner.

Two parking meters, one on the left and one on the right, stand next to each other against an urban street backdrop. The left parking meter is painted blue with a silver

A photo of two parking meters, one vibrant yellow and one rustic brown, standing side by side in a bustling city street, marked by their distinct colors and designs.

A photo of two parking meters with a rusty green finish and a metal surface, featuring a prominent circular design with digital displays showing time limits, side-by-side at the

Task: Color_Attr
Prompt: a photo of a blue clock and a white cup.

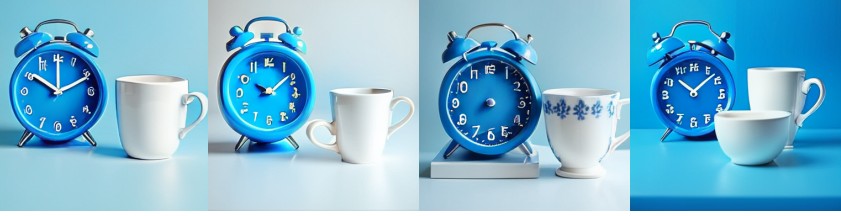

A vibrant blue clock with white clock hands showcases a bright white cup beside it, highlighting the contrast between the two objects.

A captivating blue clock with a rounded shape features a white cup resting nearby, symbolizing the timeless bond between morning routines and a calm, refreshing moment.

A blue clock with white numbers sits atop a pedestal and beside a pristine white cup with blue patterns.

A serene blue clock with a pristine white cup below, showcasing a harmonious balance between time and refreshment.

Task: Position
Prompt: a photo of a train above a potted plant.

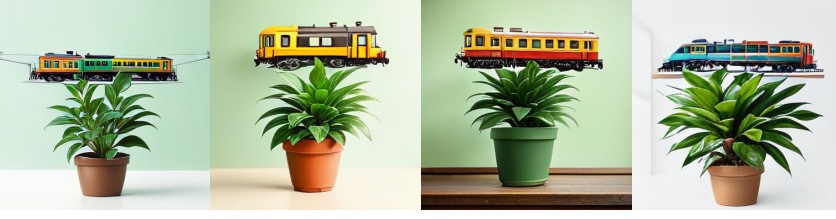

The image shows a train above a potted plant, displaying a scenic transportation scene with a vibrant, colorful train and a healthy, green potted plant below.

A vibrant, yellow-colored train hovers above a lush, green potted plant, showcasing a sense of harmony between man-made and natural elements.

A vintage-style train with yellow and red accents sits above a green artificial potted plant, contrasting sharply with the natural background.

A vibrant train depicted at an elevated platform above a lush green potted plant, highlighting the harmony between transportation and nature.

Figure 7: Uncurated Results from GenEval across different subtasks. Top 2 rows are from Cap Length and the bottom 2 are from Target Length. The generated CoTs are below each image.

## A.4 OTHER POTENTIAL FUTURE DIRECTIONS

Other potential future directions include: (1) enlarging training set to make the model generalize to real-world user prompts, (2) further improve the results with test-time scaling over seeds. In addition, this work leaves multi-step image generation with CoT and its efficiency to future work. Currently CoT for autoregressive image generation focuses on a single-turn image generation where the model reasons both language tokens and image tokens (Jiang et al., 2025). However, one can also decompose an image generation task (such as "a ball and a party hat") into multiple steps, where the model reasoning about the number of steps needed and what each step needs to generate or edit, and self-criticism (e.g. generate a ball first, then a party hat).

