# OpenReview forum: "Improving Chain-of-Thought Efficiency for Autoregressive Image Generation"
_ICLR.cc/2026/Conference — ICLR 2026 Conference Withdrawn Submission_

### Official Review · Reviewer_zzn8 · 2025-10-25

**Soundness:** 2
**Presentation:** 3
**Contribution:** 2
**Rating:** 2
**Confidence:** 3

**Summary:**

This paper proposes ShortCoTI, a reinforcement learning (RL) optimization framework designed to address the issue of Visual Overthinking in autoregressive image generation models. This phenomenon manifests as computational redundancy and potential error introduction in the Chain-of-Thought (CoT) reasoning sequences. The core contribution of ShortCoTI lies in designing a dynamically adaptive length penalty term, which is integrated into the baseline model’s reward function. This approach dynamically generates more concise CoT sequences, successfully reducing the CoT token length by half while maintaining or improving image-text alignment accuracy, and enhancing overall generation efficiency. Experimental results demonstrate that ShortCoTI improves efficiency and also mitigates visual hallucinations in the T2I-R1 model by eliminating redundant information.

**Strengths:**

1. This work identifies the key problem of "Visual Overthinking" and successfully reduces the redundant CoT token length by approximately 54%, improving reasoning efficiency.

2. The dynamically adaptive length penalty mechanism introduced by ShortCoTI is flexible than fixed truncation or linear penalties.

3. Experimental results strongly demonstrate that the improvement in efficiency does not come at the cost of generation quality; instead, it enhances image-text alignment accuracy.

**Weaknesses:**

1. All the work in this paper is built upon the T2I-R1 architecture and primarily aims to reduce redundancy in textual CoT. However, this method lacks generalizability for other generative models that do not follow this paradigm.

2. Simply adding a length penalty to the reward function is naive, as it cannot guarantee the accuracy of the shortened CoT or its proper alignment with the original prompt. There is a lack of in-depth analysis of error types. Does ShortCoTI mainly address specific types of CoT errors (such as counting errors or object hallucinations) ? The authors need to provide more clarification.

3. Although the paper mentions that the textual CoT reasoning time is reduced by half, the reasoning time at this stage does not constitute the major part of the overall image generation process. Therefore, the improvement in end-to-end inference efficiency is limited.

**Questions:**

See Weaknesses.

---

### Official Review · Reviewer_W8GL · 2025-10-29

**Soundness:** 3
**Presentation:** 3
**Contribution:** 2
**Rating:** 4
**Confidence:** 4

**Summary:**

This paper introduces **ShortCoTI**, a reinforcement learning method designed to optimize Chain-of-Thought (CoT) reasoning length in autoregressive text-to-image generation. specifically, the author design a difficulty-adaptive length penalty within the GRPO framework, where the penalty weight is modulated by the estimated difficulty of each prompt, derived from multi-model rewards. Experiments on GenEval and T2I-CompBench benchmarks show that ShortCoTI achieves roughly 54% shorter CoTs while maintaining or slightly improving alignment and aesthetic scores.

**Strengths:**

- The paper is well-motivated and addresses a practical issue. Reducing redundant CoT in image generation directly improves efficiency and reduces hallucination.

- The approach, Adding a soft penalty on CoT length to the T2I-R1 reward function is intuitive. Experiments on GenEval and T2I-CompBench benchmarks demonstrate that **ShortCoTI** achieves shorter CoTs while maintaining or slightly improving alignment and aesthetic scores.

**Weaknesses:**

- The paper is technically sound and addresses a practical problem in CoT-based T2I generation. However, my biggest concern is its novelty. It is more like an empirical experimental report based on the T2I-R1. The proposed solution, a difficulty-adaptive length penalty within an existing GRPO framework,  is primarily an engineering refinement rather than a fundamentally new algorithmic contribution. Similar concepts have been explored in LLM CoT efficiency studies.

- The experiments lack of human evaluation and failure case analysis. It is highly recommended that the authors include a small-scale human preference study (e.g., 200 prompts) to validate that automated aesthetic and alignment metrics align with human perception. It would also be beneficial to include an analysis of failure cases.

- The paper should do quantitative ablations on the  choice of alpha and L_T in eq. (4).

**Questions:**

See the weakness.

---

### Official Review · Reviewer_acMp · 2025-11-01

**Soundness:** 3
**Presentation:** 3
**Contribution:** 3
**Rating:** 6
**Confidence:** 2

**Summary:**

This paper addresses the "visual overthinking" problem in autoregressive text-to-image models that use chain-of-thought (CoT) reasoning. The authors observe that models like T2I-R1 often generate unnecessarily verbose reasoning prompts that increase computational costs and can introduce contradictory details. They propose ShortCoTI, a reinforcement learning framework based on Group Relative Policy Optimization (GRPO) that incorporates an adaptive length penalty scaled by task difficulty. The method reduces CoT length by 54% while slightly improving generation quality (+1.44% on T2I-CompBench, +2.76% on GenEval). The paper also explores simpler alternatives (Cap Length, Target Length) and provides analysis on when CoT is beneficial.

**Strengths:**

- Clear Problem Identification: The authors identify and quantify "visual overthinking" in autoregressive image generation, with compelling examples showing how verbose CoT can harm quality.

- Adaptive Solution: The difficulty-aware length penalty is a simple yet effective solution that preserves CoT benefits for complex prompts while trimming redundancy for simple ones.

- Practical Impact: The proposed method achieved end-to-end inference speedup with quality improvements, making this immediately useful.

**Weaknesses:**

- It would be better if the authors could apply their proposed method to other baselines, rather than T2I-R1 alone, which could help strengthen its generalizability.

- It would be better if the authors could discuss some failure cases or demonstrate cases with complex real-user prompts.

- As seen in Table 2, despite achieving a substantial reduction in CoT length, the end-to-end inference time improvement is modest (from 34.85s to 32.01s).

**Questions:**

None

---

### Note · Authors · 2025-11-13

I have read and agree with the venue's withdrawal policy on behalf of myself and my co-authors.